# Drug-Induced Liver Injury—Pharmacological Spectrum Among Children

**DOI:** 10.3390/ijms26052006

**Published:** 2025-02-25

**Authors:** Bianca Raluca Maris, Alina Grama, Tudor Lucian Pop

**Affiliations:** 12nd Pediatric Discipline, Department of Mother and Child, Faculty of Medicine, “Iuliu Hațieganu” University of Medicine and Pharmacy, 400012 Cluj-Napoca, Romania; mateescu_bianca_raluca@elearn.umfcluj.ro (B.R.M.); tudor.pop@umfcluj.ro (T.L.P.); 22nd Pediatric Clinic, Emergency Clinical Hospital for Children, 400177 Cluj-Napoca, Romania

**Keywords:** toxicity, liver injury, drug metabolism, idiosyncrasy, pediatric

## Abstract

Drug-induced liver injury (DILI) is one of the main causes of acute liver failure in children. Its incidence is probably underestimated, as specific diagnostic tools are currently lacking. Over 1000 known drugs cause DILI, and the list is expanding. The aim of this review is to describe DILI pathogenesis and emphasize the drugs accountable for child DILI in order to aid its recognition. Intrinsic DILI is well described in terms of mechanism, incriminated drugs, and toxic dose. Conversely, idiosyncratic DILI (iDILI) is unpredictable, occurring as a result of a particular response to drug administration, and its occurrence cannot be foreseen in clinical studies. Half of pediatric iDILI cases are linked to antibiotics, mostly amoxicillin–clavulanate, in the immune-allergic group, while autoimmune DILI is the hallmark of minocycline and nitrofurantoin. Secondly, antiepileptics are responsible for 20% of pediatric iDILI cases, children being more prone to iDILI caused by these agents than adults. A similar tendency was observed in anti-tuberculosis drugs, higher incidences being reported in children below three years old. Current data show growing cases of iDILI related to antineoplastic agents, atomoxetine, and albendazole, so that it is advisable for clinicians to maintain a high index of suspicion regarding iDILI.

## 1. Introduction

Drug-induced liver injury (DILI) is a disease of ongoing global research, as it currently is the main reason for drug market withdrawal [1]. Moreover, DILI is one of the leading causes of acute liver failure (ALF) in Western countries. Its idiosyncratic subtype is responsible for about 10% of ALF cases in the United States, surpassing viral hepatitis [2,3,4]. However, its true incidence is difficult to assess, because specific, readily available laboratory studies are lacking, and DILI is most probably underdiagnosed [5]. There is an extensive list of the drugs most commonly involved in DILI pathogenesis, but more are being discovered during clinical practice [6]. To this date, over 1000 drugs have been proven to cause DILI [3].

There are two main types of DILI: intrinsic and idiosyncratic. Intrinsic DILI is dose-dependent and predictable, the drug itself directly causing liver injury. The most typical medication, frequently used to exemplify the intrinsic mechanism of DILI, is acetaminophen (APAP), widely available in over-the-counter (OTC) products. Idiosyncratic DILI (iDILI), however, is unpredictable, is less dependent on the administered dose, and its development relies on individual susceptibility [7]. Multiple pharmacological agents are included in this category, including antibiotics, antiepileptic agents, nonsteroidal anti-inflammatory drugs (NSAIDs), and many others [6].

iDILI has been shown to be species-specific and associated with certain HLA polymorphisms [3]. Therefore, iDILI cannot always be foreseen in preclinical-stage drug trials, as current data suggest that less than 1% of drugs proven to cause iDILI were initially labeled as potentially liver-toxic [8]. Furthermore, clinical trials conducted on limited groups may not be able to link a certain drug to iDILI. Consequently, it sometimes is the clinician’s task to identify new incriminating drugs during the post-marketing phase [9,10]. However, specific diagnostic markers and treatment options are lacking, so research in this field is imperative [6].

Most studies that have been published on DILI are conducted on adult patients, while extensive data regarding pediatric DILI are insufficient, even though around 10% of all cases occur in children [11]. The purpose of this literature review is to emphasize the importance of timely iDILI diagnosis and assess the pharmacological spectrum in children and current needs on this subject.

## 2. Pathogenesis and DILI Subtypes

### 2.1. Drug Pharmacokinetics

After a certain drug is ingested, the pathway toward its pharmacological effect and, eventually, its elimination is included in the term “pharmacokinetics”. This concept includes four main stages, namely absorption, distribution, metabolism, and elimination [12]. The stage of drug metabolism is of particular interest when discussing DILI, as its headquarters lie within the liver [13].

The purpose of hepatic metabolism is drug inactivation and transformation into a hydrophilic form that is easily excreted in the urine. This process is divided into three chemical reaction subtypes (Figure 1) [14].

Phase I reactions are typically catalyzed by the cytochrome P oxidase superfamily (CYP450), located on the membrane of the endoplasmic reticulum (ER). They include oxidation, reduction, hydrolysis, and cyclization, which predominantly inactivate the pharmacological agent [15]. Nonetheless, some prodrugs (e.g., codeine, enalapril) are converted into their active form following phase I reactions [16,17]. Phase II reactions further inactivate the compound by conjugation under the effect of multiple enzymes, such as UDP-glucuronosyltransferase, glutathione S-transferase, aryl sulfatase, and others. Even though many drugs are first metabolized by phase I reactions and then by phase II conjugation enzymes, it is not uncommon for some pharmacological agents to initially encounter phase II reactions, so these chemical reaction subgroups are not necessarily sequential [14]. Finally, phase III reactions are involved in drug efflux from the cells, either into the biliary tract or into the sinusoidal capillaries and further into the bloodstream upon elimination [15].

Developmental pharmacokinetic studies proved that drug metabolism in children varies with age and differs from that of adults in both phase I and phase II reactions. However, these differences have not been fully discovered, as child experimental research is limited on ethical grounds [18]. As a result, many pediatric dosages are derived from adult posology and adjusted according to body weight or surface. These calculations could expose patients to sub- or supra-therapeutic doses, leading to disease progression or toxicity [19]. Advancements in the knowledge of liver metabolism ontogeny have generated physiologically-based pharmacokinetic models that integrate enzyme trends during childhood with adult data, but they cannot fully replace clinical studies [20].

In general, children require higher doses per kilogram compared with adults because of a higher body water percentage with wider drug volume distribution. Moreover, some drugs suffer extensive first hepatic passage inactivation due to a higher liver mass to body mass ratio in children [21].

Regarding drug metabolism, the main differences in children have been linked to phase I, CYP-mediated reactions. For instance, *CYP3A4* is the major isoenzyme of the CYP superfamily in the adult liver and is responsible for the metabolization of most xenobiotics. At birth, its activity is at a third of adult levels; it slowly increases during the first years of life and then surpasses adult activity until puberty. As a result, infants have a lower capacity to metabolize drugs, while that of young children is higher than in adults. As a consequence, pediatric patients require either higher or lower weight-dependent doses compared with the adult population. Other enzymes, such as *CYP2C9* and *CYP1A2*, follow the same pattern during child growth [22,23]. On the other hand, *CYP3A7* is predominant in the fetal and infant liver, but it progressively decreases until its disappearance in adults. Compared with *CYP3A4*, it is less efficient in drug metabolization [24]. Studies on phase 2 enzymes showed that glucuronidation is deficient in infants and small children until after 1–3 years of age, depending on the isoenzyme. Conversely, the sulfation pathway is mature at birth and can compensate for glucuronosyltransferase activity [19].

### 2.2. Pathogenesis of DILI Subtypes

#### 2.2.1. Intrinsic DILI

Intrinsic DILI is predictable and is frequently reproducible in animal studies. Historically, drugs that were proved to be directly hepatotoxic at low doses, such as carbon tetrachloride or chloroform, have been banned from marketing. An important characteristic of this subtype of DILI is that it mostly occurs at high doses of approved drugs, above the therapeutical range (e.g., APAP), or at well-studied, known doses (e.g., amiodarone, statins, valproate) [15,25]. However, recent data have shown that certain characteristics can lower toxicity thresholds to therapeutic doses, the most notable being inflammation caused by alcohol consumption, which enhances gut permeability to commensal bacterial products and viral hepatitis [26].

The most common manifestation is acute hepatitis, which is the result of multiple hepatocyte-interrelated abnormalities: oxidative stress, reactive oxygen species generation, mitochondrial dysfunction, and bile acid transport inhibition [27].

In the setting of common APAP doses, most of the drug undergoes glucuronic or sulfate conjugation, the resulting inactive metabolites being eliminated by the kidneys [28]. A small remaining part is converted to N-acetyl-p-benzoquinone imine (NAPQI) via phase I reactions mediated by *CYP3A4*, *CYP1A2*, and *CYP2E1* [29]. NAPQI is further conjugated with glutathione (GSH) and excreted in the urine. However, during an overdose, conjugation enzymes are oversaturated, leading to high concentrations of NAPQI, GSH depletion, and hepatocyte death (Figure 2). The mechanism of cell destruction is thought to be related to protein adduct formation as a consequence of a covalent, irreversible binding of NAPQI to mitochondrial proteins [30,31]. These adducts bring about mitochondrial dysfunction, oxidative stress, DNA fragmentation and, finally, cell necrosis [32].

#### 2.2.2. Idiosyncratic DILI

Idiosyncratic drug reactions (IDRs) occur rarely and are responsible for less than 10% of all adverse reactions [33]. The term “idiosyncratic” is defined as an individual hypersensitiveness to a certain drug or food; IDRs are adverse reactions that are not related to the pharmacodynamic effect of a drug, but rather to an unpredictable genetic predisposition [34,35]. Their incidence is not higher with increased doses, they have a variable latency period, depending on drug type, and they recur after rechallenge, commonly with a shorter latent period and more severe course, even if the initial injury was relatively mild [6,35]. Therefore, it is difficult to foresee these adverse events, so that their occurrence has led to market withdrawal of several therapeutic agents [35].

Targeted organs are, firstly, the liver, because of its crucial implication in drug metabolism, along with the skin, blood cells, bone marrow, and central nervous system. IDRs cause a wide variety of clinical manifestations, from iDILI to Stevens–Johnson syndrome (SJS), drug reaction with eosinophilia and systemic symptoms (DRESS), cytopenia, and others [36]. iDILI is classified into two mechanistic subtypes: immune-mediated and metabolic [15].

##### Immune-Mediated Idiosyncratic DILI

The liver is an immunological organ, having the role of maintaining a balanced immune response to avoid its own destruction due to massive non-self-antigen passage (e.g., food antigens) and to prevent chronic infections or neoplasm formation associated with excessive immune tolerance [37]. iDILI is mainly immune mediated, with two subtypes: immune-allergic and autoimmune [38]. As the mechanism of liver destruction is immune mediated and sometimes implies antibody production, signs of iDILI manifest only after a variable latency period, which is different with each drug, ranging from weeks to months [3,6].

Even though it is stated that iDILI is dose-independent, it has been proven that it rarely occurs with drugs administered at a daily dose of less than 10 mg [32]. Drugs administered at daily doses of over 50 mg are associated with a significantly higher risk of iDILI. Other drug-related risk factors for iDILI development are drug lipophilicity and extensive liver metabolization [15].

Histologically, the liver contains a high density of antigen-presenting cells (APCs), which are in close contact with the blood flow in the sinusoid capillaries. Those resident APCs express major histocompatibility complex (MHC) type I and II molecules, which are exposed to circulating lymphocytes and can trigger an immune reaction [37].

The drug itself or its metabolites are small molecules (haptens) that cannot elicit an immune response by themselves. Bound by cellular proteins (in this case, frequently CYP450), they form non-self-antigens, which are detected by the MHC molecules on the APCs. Subsequently, anti-drug antibody production is stimulated, while the immunological damage to liver cells is initiated and further perpetuated through several antibodies directed against self-antigens (anti-cytochrome, anti-nuclear antibodies). Moreover, the resulting cell necrosis enhances cytokine and reactive oxygen species (ROS) release, mitochondrial dysfunction, and ER stress, which further promote liver injury [3,7,38].

The immunological pattern of liver injury is either a type I or type IV hypersensitivity reaction [39]. Typical symptoms of immune-allergic iDILI are fever, rash, and lymphadenopathy, and the most common associated laboratory finding is eosinophilia. The onset of symptoms can be rapid, within hours or days of exposure, in type I IgE-mediated hypersensitivity reactions, while type IV reactions have a longer latency of a few weeks or months [40,41]. The fact that previous drug allergies represent risk factors for DILI supports the previously mentioned pathophysiology of iDILI [6]. Anticonvulsants (phenytoin, carbamazepine) and antibiotics (trimethoprim–sulfamethoxazole, cefazolin, ciprofloxacin, isoniazid) are drugs cited to cause immune-allergic iDILI [7,42].

Autoimmune DILI occurs due to a delayed drug reaction, with a longer latency, up to several years, and autoantibody production [40,43]. The evidence of autoimmunity can interfere with the diagnosis, as antibodies commonly found in autoimmune hepatitis (AIH), such as antinuclear antibodies (ANA) and anti-smooth muscle antibodies (SMA), can be positive in iDILI and are frequently associated with high total immunoglobulin G titers. Moreover, it is well known that drugs can be potential triggers of true AIH [44,45]. Several autoantibodies, including ANA, can be positive in any other type of acute liver injury, such as viral hepatitis, and are also seen in otherwise healthy people, even though this is rarely the case in children [46,47]. Therefore, differentiating between DILI and AIH can be difficult in the acute phase. Examples of pharmacological agents involved in autoimmune iDILI range from antibiotics (nitrofurantoin, amoxicillin–clavulanate, minocycline), statins (atorvastatin, simvastatin, rosuvastatin), anti-TNFα agents (Infliximab, Adalimumab), and anti-hypertensive drugs (hydralazine, methyl-dopa). The great majority of child cases are associated with antibiotics, the other drug classes being rarely administered in pediatrics compared with adult medicine [45,48].

##### Metabolic Idiosyncratic DILI

A second, non-immune subtype of idiosyncratic DILI implies an aberrant metabolism of the drug [49]. It has been proposed that polymorphisms associated with an abnormal drug biotransformation would lead to the accumulation of drug/intermediate metabolites that are presumably harmful to cell function when a certain threshold is reached (e.g., liver injury caused by isoniazid in slow-acetylators) [50]. Examples of such cell disturbances are mitochondrial dysfunction, ER stress, and inhibition of bile salt export pumps (BSEPs), which cause hepatocyte destruction and the release of intracellular components [7,49,51].

Even though it has been thought that metabolic DILI is a separate, completely independent subtype of iDILI, recent studies have shown that it most probably represents the starting point in the initiation of the forementioned immune responses that are associated with allergic and autoimmune DILI [51,52]. Therefore, several authors suggest that the distinct term of “metabolic iDILI” be cautiously adopted [4,49].

However, valproic acid (VPA) remains the prototype of this mechanism, as it seems that it does not elicit any significant activation of the immune system. Pharmacological studies have emphasized that VPA may cause iDILI through a multitude of metabolic pathways, with three main clinical manifestations: ALF with encephalopathy, hyperammonemia, and acute Reye-like syndrome [4,53]. One mechanism of VPA metabolism is beta-oxidation within the mitochondrial respiratory chain, its entrance into the mitochondria being mediated by carnitine [54]. Thus, it blocks the electron transport chain, with resulting reactive oxygen species (ROS), oxidative stress, and hepatocyte necrosis. It also inhibits fatty acid beta-oxidation, leading to microvesicular steatosis. On the other hand, it depletes carnitine reservoirs and induces hyperammonemia [53,54]. The latter is particularly important in children with underlying metabolic diseases, such as urea cycle enzyme deficiencies and hereditary carnitine deficiency, who are prone to hyperammonemia [55].

As a conclusion, given the multiple complex mechanisms (Table 1) that have been proposed, the precise pathophysiology of iDILI remains uncertain [4]. Furthermore, asserting a certain drug to a single mechanism is challenging, particularly since overlap is believed to occur frequently [15].

## 3. Diagnosis and Management

The clinical presentation of DILI is generally similar in adults and children. It manifests like other causes of acute hepatitis and usually consists of non-specific symptoms such as malaise, nausea, anorexia, abdominal pain, and low-grade fever [41,57]. However, most patients are asymptomatic, or some may present with signs of cirrhosis, if the offending drug is used chronically [6,41,58]. However, this is a rare occurrence in children [29]. Patients may also present with jaundice, acholic stools, or hyperchromic urine, or they can manifest signs of ALF (hemorrhages, encephalopathy) [41]. Signs of hypersensitivity, such as fever, lymphadenopathy, rash, or SJS, are suggestive of iDILI and have been cited to predict a better outcome in children [29,41,43,59]. Laboratory findings are represented by high transaminase levels (alanine aminotransferase—ALT, aspartate aminotransferase—AST), sometimes associated with cholestasis, direct hyperbilirubinemia, and signs of liver failure such as hypoalbuminemia and abnormal coagulation tests [40]. In the case of ALF, various scores, such as PELD, have been proven useful in the prediction of liver transplantation (LT) necessity in children [60]. Liver test alterations are considered significant for the diagnosis of DILI if ALT is over three times the upper limit of normal (ULN) and/or alkaline phosphatase (ALP) is over two times ULN [40]. As these findings are not specific to DILI, the diagnosis strongly relies on the recognition of drug consumption and establishing a link to symptom development, especially in iDILI, as intrinsic toxicity implies an overdose that can be more easily identified during anamnesis [59]. Serum levels of APAP can also be measured in some hospital settings [61].

APAP overdose has a well-described clinical and paraclinical course, depending on the elapsed time since drug consumption [30,62]. Toxic doses are also commonly accepted and validated: over 200 mg/kg or 10 g [62,63].

The clinical picture in iDILI is not as precise as that described in intrinsic DILI, as it involves a wide variety of drugs with different latencies of liver injury induction. Moreover, neither signs and symptoms nor laboratory studies are specific for iDILI, so that establishing the diagnosis can be a real challenge. It is important to emphasize that iDILI remains a diagnosis of exclusion [4,64].

Three main patterns of liver disease have been associated with iDILI, and the classification is performed by calculating the R-value, which is the ratio between ALT and ALP levels, in terms of “times above ULN”. iDILI is considered hepatocellular if R is above 5, mixed if R is between 2 and 5, and cholestatic if the ratio is below 2 [6,25,40]. Each subtype comprises acute and chronic injury types [6,58]. In children, the injury pattern is mainly hepatocellular in over 70% of cases [6,57]. Some diagnostic tools have been proposed, the Roussel Uclaf Causality Assessment Method (RUCAM) being widely accepted. It has different criteria depending on R-value and includes time relation between ingestion and disease onset, course of disease after drug cessation, the presence/absence of other risk factors for liver disease, exclusion of other causes, simultaneous use of other drugs, literature data on the presumably offending drug, and response to rechallenge [65]. However, clinicians should not rely solely on RUCAM for diagnosis, as it has not been extensively validated. Moreover, its use in pediatrics has been extrapolated from adult care [6].

iDILI diagnosis is a field of ongoing research, as novel biomarkers that specifically identify DILI and exclude other causes of liver injury have not been integrated into practice in adults. Examples of such molecules are miRNA-122, total keratin 18, and glutamate dehydrogenase [6,66]. The majority of the available studies were conducted on adults, so data in child iDILI are even scarcer. As a result, many cases remain undiagnosed, and due to its presumably low incidence and the wide variety of culprit drugs, relevant analyses are difficult to obtain [6,29]. Moreover, animal hepatotoxicity models do not seem to correlate with human clinically relevant iDILI, which further limits research on this subject. As a consequence, current studies are focused on in vitro human hepatocyte cell cultures [67].

Another promising field in iDILI diagnosis is that of pharmacogenomics. Multiple HLA and non-HLA polymorphisms have been linked to a predisposition toward iDILI in both adults and children, depending on ethnicity and the evaluated drug. The first reports were related to amoxicillin–clavulanate, in which *HLA-DRB1*15:02* is associated with a high risk of liver injury. Other drugs that have proven genetic susceptibility are trimethoprim–sulfamethoxazole, nitrofurantoin, minocycline, and carbamazepine [68]. The clinical utility of these polymorphisms is still uncertain, as systematic testing has been validated only for abacavir, an antiretroviral drug [69].

Another valuable method that assists the diagnostic process is liver biopsy. The main contribution of histopathological exams is the exclusion of other diagnoses, such as autoimmune hepatitis [6]. However, liver biopsy is not mandatory for the diagnosis of DILI, and it should be reserved for certain cases, given that it displays the potential to modify the treatment approach or to give valuable information about prognosis [70].

Apart from APAP toxicity, in which N-acetylcysteine (NAC) is an approved antidote, DILI management mainly consists of cessation of the offending drug. In iDILI, on the other hand, a placebo-controlled clinical trial showed a lower LT-free survival in pediatric ALF cases receiving NAC [6,71]. In immune-mediated iDILI cases, corticosteroids can be considered, but the evidence is low. Other interventions, such as ursodeoxycholic acid in cholestasis, S-adenosyl methionine, and carnitine have been proposed. Still, no well-conducted clinical trials have proven their efficacy and evidence derived from case reports and expert opinions, so further clinical trials are pending [6,72].

## 4. DILI in Children

DILI is a rare entity among children. Nonetheless, it is essential to be aware of the fact that the pediatric group is poorly represented in studies in general, so that a great proportion of DILI cases in children come from single-case presentations or small case series reports [57]. Moreover, some drugs documented in adult patients are not approved for child use [6]. As a result, it is highly probable that pediatric DILI is underdiagnosed, so it should not be overlooked as a possible cause of liver injury in pediatrics, especially since it accounts for about 20% of cases of child ALF [29].

### 4.1. Acetaminophen

In APAP poisoning, the most representative example for intrinsic DILI, the clinical course consists of four phases [30,62]:Phase 1: within first 24 h patients are either asymptomatic or present with nausea, vomiting, abdominal pain (normal liver tests—transaminase levels begin to rise after 12 h with massive doses).Phase 2: 24–72 h after ingestion, patients present right upper quadrant pain (acute hepatitis—elevated liver enzymes, coagulopathy, or renal dysfunction may arise).Phase 3: 72–96 h after ingestion patients present jaundice, coagulopathy, encephalopathy, oliguria, edema (peak of liver dysfunction—acute liver failure, kidney failure, multi-organ failure, death).Phase 4: beginning with day 4, up to 2 weeks after ingestion, is recovery (delayed histological healing—up to 3 months).

Acetaminophen-associated life-threatening DILI usually occurs with intentional overdoses, teenagers and young adults being the most frequently affected [61,73]. However, accidental poisoning may occur, especially in infants and children, with dose miscalculations or repetitive supra-therapeutic doses that are below the commonly accepted toxic dosage [74,75]. The risk of acute hepatitis as a result of APAP toxicity is dose-related, as has been already stated. Single ingestions of over 200 mg/kg or 10 g (whichever is less) are considered to have a high probability of causing DILI, even though some believe that a lower threshold of 150 mg/kg or 7.5 g/day is valid [62,63].

APAP toxicity been reported as the most common cause of hospitalization associated with drug poisoning for self-harm purposes in the United States [76]. APAP poisonings can be divided into two separate subtypes with regards to the motive of drug ingestion: intentional and unintentional [77].

Alander et al. studied the risk factors associated with hepatotoxicity in children presenting with APAP overdose. They identified 322 cases, 53.4% of which were unintentional ingestions and consisted of unsupervised infants or small children that had access to medication bottles, with a median age of 2 years old in this group. The median dose was 150 mg/kg, with 8.1% receiving antidote treatment with N-acetylcysteine and only 0.6% developing acute hepatitis, with no cases of acute liver failure [77].

Interestingly, several studies have concluded that pediatric APAP overdose is associated with a lower rate of hepatotoxicity and ALF compared with adult cases. Explanations vary from a higher probability of post-ingestion vomiting, early emergency-room presentation, and low rate of coingestions (e.g., alcohol) to lower NAPQI-generation capacity, increased glutathione regeneration rates, and greater sulfate-conjugation capacity [6,78,79].

Part of the burden of accidental APAP poisonings could be carried by uninformed caregivers who are unaware of adequate dosages, time between doses, and potential toxicity. Moreover, caregivers may not know that certain analgesics/antipyretics contain APAP, so that they concomitantly administer various commercially available preparations of the same pharmacological agent [80]. It is important to mention that the preferred formulation of APAP is that of oral solutions (e.g., syrups) because of precise dosing, while suppositories carry a greater risk of toxicity due to fixed concentrations and variable absorption [80,81].

On the other hand, intentional APAP overdoses are linked to higher rates of acute hepatitis and ALF. Alander et al. noted that 43.5% of APAP-related presentations were due to suicide attempts, the median age being 14 years old and girls being more frequently affected. Median doses were slightly higher than those in the unintentional group, 170 mg/kg [77]. A significantly greater proportion of intentional overdoses was reported more than 15 years later, 89.5% of the 9935 cases of APAP ingestions registered in 2016 in the USA being voluntary and mostly associated with psychiatric disorders such as depression and anxiety [76].

Finally, risk factors for acute hepatitis related to APAP ingestions consist of late presentation (beyond first 24 h), age (adolescents), dose (>150 mg/kg) and self-harm intentions [77].

### 4.2. Antibiotics

Antibiotics are frequently accountable for iDILI, in around 50% of cases, more recent studies reporting even higher rates [82,83]. It is understandable that they represent the leading cause of iDILI, as they are widely used, with various indications and multiple available formulations. The most frequently used antibiotics in children are beta-lactams, with two strongly represented agents: amoxicillin and cefuroxime [84,85].

The mean onset time since drug consumption is 20–30 days in the case of amoxicillin–clavulanate (AC). The most frequent complaint is jaundice, while the most encountered pattern of disease is mixed (hepatocellular/cholestatic). Histological findings in adults show evidence of an immune-allergic mechanism, while autoantibodies are seldom positive [86,87]. Several studies have shown associations between several HLA haplotypes and iDILI risk after AC use, while others have debated which of the two substances should be blamed for liver toxicity. Interestingly, there are few cases of iDILI reported after amoxicillin alone or other aminopenicillins, such as ampicillin, and the disease course has been described as merely different, of lower severity [88,89]. Petrov et al. found that clavulanic acid is the trigger of liver injury, especially of cholestasis, as it inhibits bile acid transporters and promotes oxidative stress [90]. AC was the most encountered drug in children with iDILI, in 31% of cases in a prospective observational study, even though other authors reported a lower incidence [91]. On the other hand, these findings are consistent with adult studies [57,75,92]. The true burden of AC in causing child iDILI could be underestimated, considering the size of the available study groups (<100 patients) and the frequent use of AC, of around 80 million annual prescriptions in the United States [93].

Anti-staphylococcal penicillins (oxypenicillins) are another class of beta-lactam antibiotics that are commonly used in pediatric practice and have been proven to cause iDILI as well [82,94]. Even though older age is a well-known risk factor for iDILI development after oxypenicillin administration, there are reports of cases in infants and children of all ages [89,95,96]. The incidence of iDILI was low in patients receiving oral oxypenicillins, such as flucloxacillin, but was found to be significantly higher following intravenous agents. For instance, Tang et al. found that one in four children treated with intravenous oxacillin were subsequently diagnosed with iDILI, mostly hepatocellular. Liver test abnormalities appeared earlier in intravenous formulations, after a mean of 9 days compared with 19 days seen in oral administration. The authors suggest liver enzymes be routinely monitored in all children receiving intravenous anti-staphylococcal penicillins [96].

Cephalosporins are generally thought to be the cause of only a few DILI cases, even though cases are rising [89,97]. Liver injury is typically cholestatic, with rare cases of ALF being cited [98]. Some cases of child iDILI caused by second-, third-, and fourth-generation cephalosporins have also been reported following antibiotic regimens with cefuroxime, cefixime, ceftriaxone, and cefepime in patients as young as six months old [99,100,101,102]. Although ceftriaxone-associated pseudolithiasis is not included under the term of DILI, it is worth mentioning, as pediatric patients face a significantly higher risk of this adverse reaction than adults. It usually occurs during the first week of treatment, while some authors suggest that over 25% of treated children will develop transient cholelithiasis. It is more frequent in patients treated with high doses administered as an intravenous bolus [89,103,104].

The combination of trimethoprim–sulfamethoxazole (TMP-STX) is a well-established example of the immune-allergic subtype of iDILI, owing to its sulfonamide component. Latency is variable, from several days to months, and disease onset can consist of signs of hypersensitivity, such as fever and rash [75,89]. Cholestasis is the primary underlying liver damage in adults, both acute and chronic, even though hepatocellular and mixed types have also been described [6,105]. Interestingly, younger age has been associated with a higher risk of allergic manifestations and a hepatocellular pattern of liver enzyme abnormalities [106]. A systematic review conducted by Burgos et al. concluded that TMP-STX is usually associated with mild elevations in liver tests [107]. However, they point out that isolated cases of severe liver outcomes are available as well. Concordantly, Shi et al. reported that ALF, LT, and death have been encountered in TMP-STX treated patients [75,107].

Nitrofurantoin (NF) is a broad-spectrum antibiotic mainly indicated for urinary tract infections in children over 1 month old. Together with minocycline, it is generally accepted that NF is the primary cause of iDILI with autoimmune features [108]. Liver injury can occur in 1 in 1500 patients using NF and can develop with either short (up to 1 month), or prolonged latency (over 1 year) with a hepatocellular pattern of liver damage [6,109]. Long-term use of NF is especially likely to induce autoimmune iDILI, with the possibility of progression to cirrhosis and LT or death, so NF should be cautiously indicated for long-term prophylaxis [110,111]. Many cases were documented after short-term use as well, so not only long-term treatment poses a risk for iDILI [112].

On the other hand, pediatric cases of iDILI with autoimmune features caused by minocycline are abundant. Belonging to the class of tetracyclines, minocycline has been proven to be beneficial in juvenile acne and many other infections, having good coverage of both gram-positive and gram-negative bacteria along with other non-antibiotic effects—anti-inflammatory, immunomodulatory, and antioxidant [113,114]. Moreover, some recent studies have also shown its potential link to neuroprotection [115,116]. However, the risk of liver injury resembling AIH is not negligible with minocycline use in adolescents, especially since treatment courses are usually prolonged to a mean of 3–4 months and can extend to years [117,118]. The DILIN Prospective Study declared minocycline as a leading cause of pediatric DILI, while the VigiBase recorded 117 such cases, thus classing minocycline in the top 10 drugs associated with this disease [57,119]. Symptoms appear with a mean latency time of about 500 days [57]. Accordingly, de Boer et al. observed that half of their patients presented with symptoms related to DILI after more than 1 year of treatment [43]. Notably, all patients in the DILIN Prospective Study and 81% of patients reported by DiPaola et al. had positive autoantibodies specific to AIH (ANA, SMA), data which are in concordance with adult trends. Even though some patients may have proof of chronic hepatitis and liver fibrosis on biopsy, most cases fully recover after drug discontinuation and short courses of corticoid therapy as needed [57,120,121].

Rare cases of macrolide-induced liver injury have been described in adults, and as it represents a frequently prescribed antibiotic class in children as well, it should not be overlooked. Moreover, one representant of this class, telithromycin, has been banned due to proven hepatotoxicity [122]. Each macrolide induces DILI through a different mechanism. While erythromycin inhibits bile acid transporters, clarithromycin induces mitochondrial dysfunction, but for azithromycin, the mechanism remains unknown [122,123]. The typical damage pattern in azithromycin-induced liver injury is hepatocellular in adult patients, with mostly mild liver enzyme elevations [124]. The DILIN prospective study identified four pediatric patients with iDILI associated with azithromycin. Two of them had a cholestatic pattern of injury, with one case of chronic DILI and ductopenia [57,124]. Similarly, DiPaola et al. identified four cases associated with pediatric azithromycin use, three of whom had proof of cholestasis on histology [120]. Notably, cases can arise even after drug withdrawal and are frequently seen in patients with preexisting liver disease [83,123].

### 4.3. Anti-Tuberculosis Drugs

Tuberculosis (TB) is a prevalent disease worldwide, especially in low- and middle-income countries, and is associated with relevant morbidity and mortality rates in children, 11% of worldwide TB cases being reported in pediatric patients [125]. TB treatment implies long-term administration, from 4 months to 1 year, of drug associations in various regimens depending on disease severity and site [126]. The cornerstone of TB management is represented by isoniazid, rifampicin, and pyrazinamide, while DILI is the most frequent adverse effect of these drugs [127]. Anti-tuberculosis drug-induced liver injury (ATLI) is a major problem as it is an important cause of drug resistance development, so it should be judiciously tackled [128,129].

Recent studies showed that children, especially below 3 years old, have a higher risk of ATLI compared with adults, despite previous reports to the contrary [75,129]. The incidence of ATLI is differently reported, between 14 and 27% [127,130]. Most cases emerge in the first two months during the intensive phase of treatment, which generally includes a combination of three or even four drugs [128,130,131]. The pattern of liver injury is typically hepatocellular [120]. Several risk factors have been identified, such as younger age, malnutrition, hypoalbuminemia, anemia, concurrent use of hepatotoxic drugs, and TB meningitis [127,128,131].

Among all anti-tuberculosis drugs, isoniazid has the greatest link to ATLI and is the only agent that was reported by the DILIN Prospective Study in children [57,130]. It is generally accepted that it causes DILI by means of metabolic derangements caused by acetylhydrazine, a phase 2 metabolite, which in turn forms protein adducts after CYP450 metabolism that bring about lipid peroxidation, oxidative stress, mitochondrial dysfunction, and DNA fragmentation [132,133]. An association between CYP450 inducers such as rifampicin and a slow-acetylator state greatly enhances the chance of ATLI. Even though rifampicin rarely causes ATLI, the standard combination of rifampicin and isoniazid is known to have significantly higher hepatotoxic potential than each of the separate drugs [75,133].

### 4.4. Antiepileptic Drugs

The second most common drug class involved in pediatric iDILI in the USA is that of antiepileptic drugs (AEDs) in about 20% of cases [120]. Some authors suggest that children are particularly prone to DILI induced by epilepsy treatment because of increased CYP450 metabolism, which generates hepatotoxic compounds. Moreover, polytherapy is frequently required, while some AEDs are potent microsomal enzyme inducers, so that drug associations impose a greater risk of iDILI [6,23]. Of the AEDs, valproate, phenytoin, and carbamazepine have been the most implicated due to the aromatic ring in their structure. However, other AEDs can cause iDILI as well [6,134].

There are three clinical syndromes of liver injury related to valproate consumption: hyperammonemia, acute hepatocellular or mixed injury, and Reye-like syndrome. Latency is different in each subtype, but it is generally over 1 month, though it can extend to several years [53]. The highest risk of fatal disease is seen in children with inborn errors of metabolism or aged below 2 years old, with an incidence of 1:600 in this age group [75]. The exact pathogenesis is difficult to establish, even though some hypotheses have been raised, such as carnitine depletion, inhibition of fatty acid beta-oxidation, or increased production of toxic metabolites such as 4-ene-VPA [23,75]. VPA is among the top five medications that are associated with DILI in pediatric patients [119,133]. Several reports have found asymptomatic elevations in ammonia levels of children treated with VPA in up to 70% of cases, while less than 5% developed clinical signs of encephalopathy. ALT, ALP, and GGT levels were only mildly elevated in most patients, so that VPA could be considered safe in young children with regard to liver side effects. However, these data come from studies conducted on small groups [135,136].

Phenytoin and carbamazepine are AEDs that have been classified as highly probable to cause iDILI. Patients may develop high GGT levels with no significant hepatic alterations, probably as a sign of liver enzyme induction. Moreover, this fact explains why co-administration of the two or association with other hepatotoxic drugs increases the risk of liver injury. They generally cause an immune-allergic type of idiosyncratic reaction, with symptoms and signs of systemic hypersensitivity, such as fever, rash, and lymphadenopathy, that precede jaundice, hyperchromic urine, or other liver-related symptoms [137,138]. This clinical pattern has been assigned the term “anticonvulsant hypersensitivity syndrome (AHS)” [6]. Symptoms usually develop up to 6 weeks after drug prescription, and the pattern of injury is hepatocellular in children compared with adults, who can also present with cholestatic or mixed-type iDILI, with cases of biliary duct paucity in the case of carbamazepine [75]. Chalasani et al. reported serious associated cutaneous reactions such as DRESS or SJS in the majority of iDILI patients, namely, in 63% in the phenytoin group and 45% in the carbamazepine-treated group. However, most patients were adults [139]. Several cases of pediatric iDILI caused by phenytoin or carbamazepine are available, 57 of the former and 104 of the latter being included in Ferrajolo’s study [119]. In a retrospective study conducted by Devarbhavi et al. on both adults and children, pediatric cases of concomitant iDILI and SJS were almost exclusively caused by aromatic antiepileptics. Of note, children had better outcomes compared with adults [140]. Nevertheless, severe, life-threatening cases have also been described, such as that of an 11-year-old girl who developed iDILI, SJS, and secondary hemophagocytic lymphohistiocytosis (HLH) as a result of carbamazepine treatment [141]. Phenobarbital-associated AHS cases have been cited as well, but with a lower frequency, the LiverTox data showing that fewer than 1% of patients develop liver enzyme abnormalities, with even rarer occasions of clinically apparent disease [119,142]. Remarkably, however, phenobarbital was found to be the most consistent associated drug in lethal pediatric cases of valproate iDILI that were undergoing polytherapy for disease control [143].

Cases of lamotrigine-induced liver injury have also been cited, while some authors suggest a 10 times higher risk of iDILI in children than in adults, being even more significant with higher doses or rapid dose escalation [134,144]. Lamotrigine belongs to the group of second-generation AEDs that has been thought to be less harmful to the liver [145]. Interestingly, the DILIN Prospective Study identified lamotrigine as the most common AED to cause pediatric iDILI [57]. It usually causes a hepatocellular type of iDILI with immune-allergic features, which can be similar to that of carbamazepine or phenytoin, and symptoms appear after a mean of one month of treatment [6,139]. Some cases of ALF caused by lamotrigine have been reported, with one death and one case of LT. Even though it is thought to occur rarely, this particular cause of iDILI should not be neglected, especially in epileptic children undergoing polytherapy [146,147,148].

The DILIN Prospective Study revealed that the incidence of iDILI caused by AEDs has significantly decreased over the last 15 years due to the preference of clinicians toward newer AEDs [139]. Even though levetiracetam and other new-generation AEDs were not initially thought to cause DILI, a newer report conducted by Petrovic et al. proved otherwise. They reported 870 iDILI cases associated with new AEDs, including levetiracetam, but also topiramate and felbamate. Moreover, severe cases were cited as well [136,139,149].

### 4.5. Nonsteroidal Anti-Inflammatory Drugs (NSAIDs)

NSAIDs account for about 10% of all iDILI cases [150]. They are largely used for their antipyretic, analgesic, and anti-inflammatory properties in both acute and chronic illnesses. It is a highly heterogenous group, but it includes pharmacological agents that inhibit COX enzymes, either selectively or non-selectively [151].

Most cases of NSAIDs-associated DILI in adults are related to diclofenac, an acetic acid derivative, or nimesulide, a sulfonanilide. They are contraindicated in children below 14 and 12 years old, respectively, so data regarding iDILI in children are lacking [150,152]. Furthermore, nimesulide has been banned in multiple countries due to hepatotoxicity risks [153]. The pattern of liver injury is hepatocellular, belonging to the metabolic subgroup, as they both primarily cause mitochondrial dysfunction [154]. Moreover, topical agents containing diclofenac have also been documented to cause liver-enzyme elevations [155].

Ibuprofen, a propionic acid derivate generally considered safe, is the most frequently used NSAID in children. Transient abnormalities in liver tests have been described, mostly following high doses, although cases of iDILI have arisen, with a relatively low incidence [150,156]. Notably, adult studies have concluded that ibuprofen-induced iDILI is frequently hepatocellular [154,157]. Zoubek et al. analyzed available data on iDILI related to ibuprofen use. In the pediatric group, the mean age at diagnosis was 10 years old, and the mean latency of symptom development was 12 days [157]. Pediatric reports have proven that the main pattern of injury is cholestatic and is often associated with SJS or toxic epidermal necrolysis (TEN), as a proof of its immune-allergic pathogenesis. Histologically, most cases revealed ductopenia, consistent with the diagnosis of vanishing bile duct syndrome (VBDS) [158,159,160,161,162].

Aspirin-associated liver injury is a distinct entity in children. Even though salicylate poisonings can occur at high doses of aspirin (over 150 mg/kg), the primary manifestations are not necessarily hepatic, and they are rarely encountered since the emergence of other NSAIDs and the strict limitation of aspirin use in children to a few indications such as Kawasaki disease. Reye’s disease, a specific type of hepatic disease caused by aspirin use in patients with viral infections such as influenza and varicella, has been acknowledged since the 1960s [163,164]. It is caused by acute mitochondrial dysfunction that is exacerbated by the high inflammatory status of the ongoing infection and leads to fatty-acid oxidation inhibition, liver microvesicular steatosis, hypoglycemia, lactic acidosis, hyperammonemia, and cerebral edema [165,166]. Few recent case reports of Reye’s disease in patients receiving anti-inflammatory doses of aspirin for Kawasaki disease are available [167,168,169]. On the other hand, low-dose, antiplatelet aspirin doses are considered safe, as no certain association with Reye’s disease has been proven. However, the general recommendation is that of aspirin cessation and substitution with a different antiplatelet agent in the case of flu or varicella outbreak [170,171,172].

### 4.6. Antineoplastic Agents

Acute lymphoblastic leukemia (ALL) is the most commonly encountered pediatric cancer, with an incidence of about 4 cases in 100,000 children, as reported in the USA [173,174]. Survival rates have increased in recent years due to advancements in treatment and reach about 90%. Chemotherapy is an indispensable tool in malignant hemopathies, long-term administration of various blocks of antineoplastic drugs being required. For example, in ALL, a total of 2–3 years of continuous chemotherapy is needed, in three main phases: induction, consolidation, and maintenance [174]. Chemotherapy agents have been attributed to a broad spectrum of liver abnormalities, from asymptomatic elevations in serum levels of transaminases to acute drug-induced hepatitis, cholestasis, steatohepatitis, pseudocirrhosis, or sinusoidal obstruction syndrome [175].

Horvath et al. analyzed 26 cases of children with ALL and observed that 77% developed liver toxicity, mostly during maintenance therapy with weekly methotrexate and daily 6-mercaptopurine. The main pattern of liver enzymes was hepatocellular. Two cases of liver fibrosis and portal hypertension were reported [176]. Urrutia-Maldonado et al. concluded that idiosyncratic liver injury is relatively frequent in pediatric cancer patients, with a high risk of relapse. They identified 22 cases, mostly in children with ALL. All patients had at least two episodes of DILI despite dose reduction, this fact being in favor of drug idiosyncrasy. Notably, in 95.9% of events, methotrexate appeared to be the offending drug. Nevertheless, precisely establishing culprit drugs is a difficult task in the setting of chemotherapy, with blocks of multiple pharmacological agents. Consistent with the findings of Horvath et al., the pattern of injury was mainly hepatocellular [177].

A retrospective study conducted by Quin et al. evaluated the risk of DILI during consolidation therapy for ALL in children. DILI was significantly more frequent in the high-risk (HR) group (5.2% of patients) compared with the low-risk or intermediate-risk (LR/IR) group (2.5% of patients). Liver injury significantly prolonged hospital stays in LR/IR patients but not in HR patients. Risk factors for liver injury were found to be younger age (< 5 years), lower baseline albumin, and multiple treatment courses (>5) in LR/IR patients, while in HR patients, lower albumin and higher initial GGT levels were associated with iDILI risk. The median onset of DILI was after 3 days of treatment initiation, and the highest probability of causing DILI was seen with high-dose methotrexate, high-dose cytarabine, and high-dose pegaspargase [178].

Ferrajolo et al. found that methotrexate, 6-mercaptoputine, and thioguanine use was associated with a relatively high probability of iDILI induction, as expressed by odds ratios (ORs) of 3.2, 4.2, and 3.9, respectively [119]. Liu et al. found that vincristine, cyclophosphamide, etoposide, and cytarabine can also be offending drugs in iDILI cases, with ORs of > 2. In their analysis, the forementioned drugs, along with methotrexate, were listed as the top ten drugs incriminated for iDILI pathogenesis [133]. Lai et al. evaluated 460 DILI cases in Chinese children and concluded that antineoplastics, closely followed by antibiotics, were the most commonly involved drugs. Methotrexate-associated iDILI was found in 62 cases, and other antineoplastics were involved in 54 cases. Over 90% of them were mild forms of disease, even though one fatality was registered in a 10-year-old boy that developed DILI-related ALF. Regular monitoring of liver tests and precocious changes in treatment regimens could be responsible for the tendency of antineoplastic DILI to be milder [179].

Methotrexate, a folate metabolism antagonist, is involved in iDILI, even though the exact mechanism is not fully understood. Long-term treatment, even in lower doses, as seen in juvenile idiopathic arthritis, has been associated with chronic iDILI in the form of liver fibrosis [180,181,182]. In vitro studies performed by Schmidt et al. proved that methotrexate activated liver stellate cells through glutathione depletion, reactive oxygen species generation, and ER stress [183]. MTHFR polymorphisms have not been linked to increased susceptibility to methotrexate-related DILI in children [184]. 6-mercaptopurine, on the other hand, induces mixed-type iDILI, with frequent cholestasis that can progress to chronicity. Idiosyncrasy has been established as the main mechanism, even though events of liver injury are more common with higher doses. Similar to methotrexate, mutations of thiopurine methyltransferase have not been associated with a higher risk of liver toxicity [185].

Chronic liver disease has been well described with many chemotherapy agents and with various histological findings. Liver fibrosis, for instance, is caused by methotrexate, but also 6-mercaptopurine and 6-thioguanine. Steatohepatitis develops after 3 to 12 months of therapy with methotrexate, asparaginase, 5-fluorouracil, and others, as a consequence of impeding fatty acid beta-oxidation. Focal nodular hyperplasia can be seen following regimens of methotrexate, 6-mercaptopurine, 6-thioguanine, or oxaliplatin, while pseudocirrhosis, a form of nonfibrotic diffuse regenerative nodular hyperplasia, can develop after methotrexate, oxaliplatin, and 5-fluorouracil [186].

Sinusoid obstruction syndrome (SOS) is a life-threatening complication of antineoplastics that can develop especially during high-dose cytoreductive therapy preceding allogenic hematopoietic cell transplantation (HCT) [187]. Up to one third of child HCT recipients present with SOS, and its incidence is higher compared with adult patients [188]. SOS implies endothelial cell injury of the sinusoidal capillaries and subsequent venous flow obstruction followed by collagen deposit formation with no initial hepatocyte abnormality [186]. Clinical signs consist of painful hepatomegaly, weight gain, encephalopathy, and jaundice [188]. Cyclophosphamide, 6-mercaptopurine, cytarabine, or oxaliplatin have been reported to cause endothelial injury and SOS after a mean of 5 weeks of treatment [186].

Immune checkpoint inhibitors (ICIs) are new antineoplastic agents that have been successful in treating adult solid tumors, but not in pediatrics, and even less in hematological cancers. ICIs have been associated with immune-mediated DILI, half of those cases developing ANA, but child data are limited [6,186].

### 4.7. Antimycotic Agents

Azoles are the most used group of systemic antimycotics, which act by fungal CYP450 inhibition. Mild and transient elevations of transaminases are well-known adverse reactions to all class representants [189,190]. The mechanism of iDILI induction is mainly unknown, even though data for BSEP and MDR3 inhibition exist for itraconazole [191]. The pattern of liver enzyme elevation is primarily cholestatic for fluconazole and itraconazole, mixed for voriconazole, and hepatocellular for ketoconazole [189]. The onset of symptoms occurs relatively close to drug consumption, during the first month, in general in the first 10 days following treatment initiation, even though longer latencies have been seldomly described [192].

Azoles are responsible for about 3% of iDILI cases [193]. According to multiple authors, voriconazole has the highest risk of iDILI among azoles, with an OR of 3.2 in children followed by itraconazole, fluconazole, and ketoconazole [133,189,192]. Existing reports suggest that approximatively one quarter of pediatric patients develop iDILI under voriconazole, while Ferrajolo et al. found that it is the sole antimycotic to be responsible for child iDILI in a statistically significant manner [119,194]. Also, the incidence of iDILI is widely variable among antifungals, patients with preexisting liver disease and the immunocompromised being more vulnerable. On the other hand, the severity of the primary fungal infection can play a part in hepatocyte dysfunction and could be a possible confounding factor [195]. An in vitro analysis by Doss et al. demonstrated high hepatotoxic effects of voriconazole in particular but also fluconazole, which were independent of dose, so that the authors concluded that especially voriconazole should be avoided in patients with concomitant liver pathology [196].

Caspofungin, a broad-spectrum antifungal belonging to the group of echinocandins, has been approved in children above the age of 3 months [197]. Elevation of transaminases is a commonly accepted reaction to caspofungin infusions, but the mechanism remains elusive. Most cases are asymptomatic, and liver function tests recover to normal after withdrawing treatment [198]. Nonetheless, caspofungin has been considered safer to administer to patients with liver disease, and even post-LT, compared with azoles. Moreover, Doss et al. found caspofungin to have the lowest hepatotoxic potential among tested antimycotic drugs. However, AST and ALT levels should be closely monitored, as some pediatric cases of iDILI were assigned to caspofungin [119,196]. Of note, echinocandins are mainly used as salvage therapies in resistant strains of Candida or Aspergillus, usually for patients with prolonged hospitalizations and significant comorbid conditions [119]. Trials conducted for caspofungin safety firstly indicated a synergic effect of cyclosporine A over the risk of liver injury. Later post-marketing studies identified no such reaction [199,200]. Koo’s retrospective review of 19 febrile neutropenic patients receiving both caspofungin and cyclosporine acknowledged 1 case of mild transaminitis with a favorable outcome despite treatment continuation. Yet previous warnings should be still considered, as most analyses were performed on small groups [201].

### 4.8. Other Drugs

#### 4.8.1. Albendazole

Albendazole, a benzimidazole, is a broad-spectrum anthelmintic agent used worldwide for the treatment of cestode, nematode, and protozoan infestations [202]. In low- and medium-income countries, it is prescribed as an annual decontamination agent [203].

The main mechanism of action is through its active metabolite, albendazole sulfoxide, which binds to β-tubulin and inhibits microtubule polymerization [204,205]. As one of the main functions of microtubules is the coordination of mitosis, along with cytoplasmatic vesicular transport and cytoskeleton structure maintenance, normal cell functioning is impossible in the presence of albendazole, and thus, the parasite dies [206,207]. It also causes the depletion of glycogen deposits and impairs ATP production in the Krebs cycle by inhibiting two of its key enzymes [204,208]. Even though all eukaryote cells contain microtubules, albendazole has been shown to be relatively selective in inhibiting their polymerization in the parasite organism, theoretically having a low affinity toward human β-tubulin [209].

Albendazole has been frequently linked to elevations in liver enzymes. However, only a few of these cases are of clinical significance and develop iDILI. Mechanistically, it usually causes an immune-mediated idiosyncratic liver injury [210]. Signs of hypersensitivity can accompany the clinical picture with autoantibody production as well. Latency is usually short, within a couple of months, with earlier occurrence after repeated treatment regimens [209].

In a case series published by Dijmarescu et al., 14 child cases of albendazole-related iDILI were reported, representing 38% of all their DILI cases. The mean time to onset was 21 days. The most commonly encountered injury pattern was hepatocellular, although cholestatic and mixed types were identified as well [211].

Recurrent albendazole-induced drug injury in pediatric patients has also been encountered. The risk of rechallenge is high, so the implicated drugs should not be further recommended after one episode of iDILI [212,213].

Dragutinovic et al. presented a pediatric case of albendazole-induced DILI with autoimmune features (high IgG titers, positive SMA) with a good outcome after drug cessation and short-term corticosteroid therapy. Albendazole can induce immune-mediated DILI, and it can also be a trigger for autoimmune hepatitis in children [214].

#### 4.8.2. Atomoxetine

Attention deficit/hyperactivity disorder (ADHD) has a global prevalence of 8% among children and adolescents. It has been considered a public health issue since the 2019 ADHD Summit [215,216].

Atomoxetine is a second-line nonstimulant drug with indication for ADHD treatment of patients above 6 years old [217]. Common adverse effects to atomoxetine include low appetite, nausea, and somnolence, but rare cases of liver toxicity have been published, with even rarer clinically manifesting iDILI [218,219]. To our knowledge, the first pediatric case of iDILI was published in 2007, 5 years after its release on the market [220]. Three other cases were later encountered, all with positive rechallenge [221]. Liver enzymes usually follow the hepatocellular pattern with onset 1 to 3 months after initiation, as has been reported in LiverTox documents [218]. Although rare, it is an indication for treatment cessation, especially since some cases of severe liver fibrosis or ALF have been reported in association with atomoxetine, with one male patient requiring LT [120,222].

The DILIN Prospective Study, however, identified atomoxetine as one of the leading agents in iDILI in children, notwithstanding the small number of pediatric cases. A significant difference was a longer latency period than that reported by LiverTox, with a median of 510 days. Notably, all cases had positive autoantibodies, either ANA alone or in association with SMA, and all of them underwent liver biopsy, with variable aspects of acute or chronic hepatitis, cholestatic hepatitis, bridging fibrosis, and eosinophil infiltrates [57]. VigiBase reports found 64 cases of atomoxetine-related DILI, with an OR of 2 [119]. Potnis et al. analyzed 8 patients retrieved from 4 case reports, the youngest patient being 8 years old. Interestingly, half of the patients developed liver enzyme abnormalities within the first month of treatment, as early as 2 days, while the other half had the onset of disease after more than 4 months, with a maximum of 699 days. These findings are consistent with both the DILIN Prospective Study and the LiverTox database and indicate two possible evolutive patterns in atomoxetine iDILI, so periodic, long-term monitoring of transaminases should be implemented [223].

#### 4.8.3. Proton Pump Inhibitors

Proton pump inhibitors (PPIs) are widely used both in adults and children for gastric acid suppression, and they act by inactivating the parietal H+/K+ ATPase, which represents the final step in gastric proton secretion [224]. They have multiple indications in children, such as gastro-esophageal reflux disease and eosinophilic esophagitis. Short courses of PPIs have been proven relatively harmless in children, with no notable adverse reactions being reported. However, long-term use can be associated with allergy, osteopenia, and infection development [225].

Notably, PPIs have been described to cause iDILI in adult patients, as several case reports included all three main substances in this class—omeprazole, esomeprazole, and pantoprazole. The pattern of injury was most commonly hepatocellular with autoimmune features [226,227,228]. Such evidence is scarce in children, with only one independent report of a child with iDILI caused by omeprazole [229]. On the other hand, a retrospective analysis identified omeprazole as a potential culprit for pediatric DILI in 828 patients, with an OR of 1.56 [230].

## 5. Conclusions

DILI should be considered in the differential diagnosis of pediatric liver injuries including ALF. Extensive lists of involved drugs are available, but new pharmacological agents are continuously added as clinical practice proves their link to DILI.

Several gaps remain in iDILI diagnosis. The RUCAM score is a valuable clinical tool, but due to limited evidence in children, it should not replace clinical judgment. Special attention should be given to the exclusion of other etiologies, especially AIH. Various biomarkers and genetic polymorphisms have been proposed for diagnosis certainty, but they have not been included in clinical guidelines. Moreover, pediatric studies on this subject are not currently available. Further research on the role of biomarkers in child iDILI is needed before clinicians can integrate such data into routine practice.

APAP is the classic example of intrinsic DILI, frequently in adolescents with suicidal intent. Antibiotics and antiepileptics have the highest probability of inducing child iDILI. However, some authors suggest that chemotherapy-related iDILI is growing in incidence. Autoimmune DILI, a distinct subtype of idiosyncratic DILI, needs judicious differentiation from AIH, as minocycline, nitrofurantoin, atomoxetine, and albendazole have been related to iDILI with autoimmune features. Clinicians should consider systematic transaminase monitoring in children receiving drugs frequently linked to iDILI, such as anti-tuberculosis, antiepileptic, and antineoplastic drugs. As no strict recommendations have been validated, the frequency and length of laboratory studies should be established according to the treatment duration and the literature-stated latency periods.

## Figures and Tables

**Figure 1 ijms-26-02006-f001:**
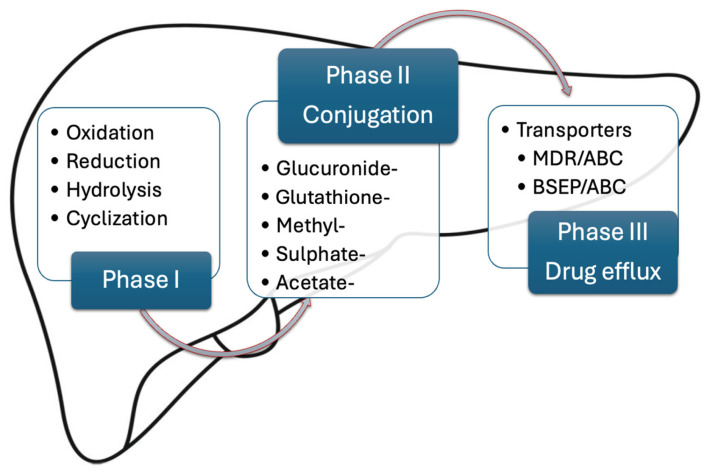
Phases of liver drug metabolism (designed after [15]). MDR/ABC= multidrug-resistant P-glycoprotein/adenosine triphosphate-binding cassette. BSEP/ABC= bile salt export protein/adenosine triphosphate-binding cassette.

**Figure 2 ijms-26-02006-f002:**
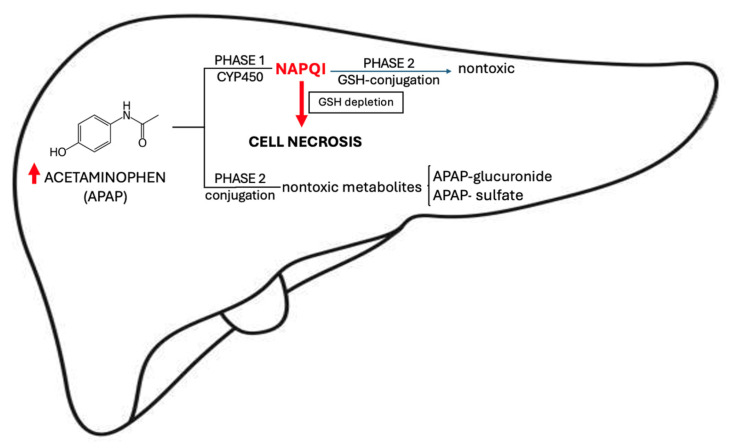
Acetaminophen metabolism (designed after [30]). APAP = acetaminophen, CYP450 = cytochrome P450, GSH = glutathione, NAPQI = N-acetyl-p-benzoquinone imine, ↑ —high.

**Table 1 ijms-26-02006-t001:** Proposed mechanisms of idiosyncratic DILI (modified after [56]).

Liver Injury	Mechanism	Drugs
Acute fatty liver	Acute mitochondrial injury Fatty acid beta-oxidation inhibition	VPA
Acute hepatic necrosis	Reactive metabolite +/− immune activation	Isoniazid, aspirin
Autoimmune-like hepatitis	Anti-drug antibodies Autoantibody production	Nitrofurantoin, minocycline, atomoxetine
Cholestatic hepatitis	Immune-mediated injury	Phenytoin, amoxicillin–clavulanate, ibuprofen
Fibrosis	Stellate cell activation/chronic endothelial cell injury	Methotrexate
Immune allergic hepatitis	Drug hypersensitivity	Trimethoprim–sulfamethoxazole, carbamazepine, phenytoin
Vanishing bile duct syndrome (VBDS)	Immune-mediated cholangiocyte injury	Amoxicillin–clavulanate, ibuprofen, sulfonamides

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
