# Peer review of "Drug-Induced Liver Injury—Pharmacological Spectrum Among Children"

_ijms, 2025, doi:10.3390/ijms26052006_

Round 1
Reviewer 1 Report
Comments and Suggestions for Authors
The authors reviewed the pathogenesis of drug-induced liver injury in pediatric patients including the drugs that are commonly involved. A special focus is on idiosyncratic drug-induced liver injury. Drugs frequently associated are antibiotics such as amoxicillin-clavulanate and anti-epileptics.
This is a review that summarizes the state-of-the-art in the field. It reads like a book chapter and there is nothing critical or new- no gap analysis, critical evaluation or any conclusions regarding future research, recommendations for clinical practice or necessary initiatives to address gaps in knowledge and/or clinical practice.
Since a critical analysis is missing, this review simply reflects the existing literature. So the review doesn't add much to the subject area compared with other published material, if anything.
Since this is a review and not a research article, the methodology is less of concern and is adequate for this type of article.
There are no real conclusions. The conclusion section is simply a brief summary without any analysis or critical evaluation.
The references are appropriate.
Author Response
Thank you very much for taking the time to review this manuscript. Please find the detailed responses below and the corresponding revisions highlighted in gray in the re-submitted files.
Point-by-point response to Comments and Suggestions for Authors:
Comments 1: This is a review that summarizes the state-of-the-art in the field. It reads like a book chapter and there is nothing critical or new- no gap analysis, critical evaluation or any conclusions regarding future research, recommendations for clinical practice or necessary initiatives to address gaps in knowledge and/or clinical practice.
Since a critical analysis is missing, this review simply reflects the existing literature. So the review doesn't add much to the subject area compared with other published material, if anything.
Response 1: Thank you for your remark. We agree with this comment, so that we have added information regarding current research in the field of iDILI diagnosis and treatment, but also limitations to incorporating them into clinical practice. You can find them in the revised manuscript at pages 8 and 9, lines 307-328 and 334-342, highlighted in gray.
Comments 2: There are no real conclusions. The conclusion section is simply a brief summary without any analysis or critical evaluation.
Response 2: Agree. We have, accordingly, revised the conclusions to emphasize current gaps and need for future research. We have added a general recommendation for clinicians regarding iDILI screening. These changes can be found in the revised manuscript at page 20, lines 893-899 and 905-909, highlighted in gray.
Reviewer 2 Report
Comments and Suggestions for Authors
Authors reviewed about DILI especially among children. This review was comprehensive and well-written. But several issues remained to be addressed in the former part. This review focus on DILI in children. But general information not but about children were shown in sections 2 and 3. In sections 2 and 3, information about children should be shown. Drug metabolism in children or immune reaction in children should be described. The changes according to growth might be preferrable.
Author Response
Thank you very much for taking the time to review this manuscript. Please find the detailed responses below and the corresponding revisions highlighted in yellow in the re-submitted files.
Point-by-point response to Comments and Suggestions for Authors
Comments 1: Authors reviewed about DILI especially among children. This review was comprehensive and well-written. But several issues remained to be addressed in the former part. This review focus on DILI in children. But general information not but about children were shown in sections 2 and 3. In sections 2 and 3, information about children should be shown. Drug metabolism in children or immune reaction in children should be described. The changes according to growth might be preferrable.
Response 1: Thank you for your remark, we found it helpful. We agree with this comment, so that we have added information regarding drug metabolism and changes in liver enzymes according to growth, with some known implications in clinical practice. You can find them in the revised manuscript at page 3, lines 89-117, highlighted in yellow.
Unfortunately, we found no data regarding immune particularities in children that are involved in iDILI pathogenesis, so that we have not included such information in the manuscript.
In section 3, we added some brief information about the clinical picture of DILI in children, as medical literature does not highlight major differences in manifestations among children, compared to adults. Additions can be found in the revised manuscript at page 7, lines 261-272, highlighted in yellow.
Round 2
Reviewer 2 Report
Comments and Suggestions for Authors
Revised manuscript was well-addressed to the reviewer's suggestion and well-written.